# Prevalence, age of decision, and interpersonal warmth judgements of childfree adults: Replication and extensions

**Jennifer Watling Neal** *, **Zachary P. Neal**

Psychology Department, Michigan State University, East Lansing, Michigan, United States of America

* jneal@msu.edu

## Abstract

Childfree individuals, who are also described as 'childless by choice' or 'voluntarily childless', have decided they do not want biological or adopted children. This is an important population to understand because its members have unique reproductive health and end-of-life needs, and they encounter challenges managing work-life balance and with stereotypes. Prior estimates of childfree adults' prevalence in the United States, their age of decision, and interpersonal warmth judgements have varied widely over time and by study design. To clarify these characteristics of the contemporary childfree population, we conduct a pre-registered direct replication of a recent population-representative study. All estimates concerning childfree adults replicate, boosting confidence in earlier conclusions that childfree people are numerous and decide early in life, and that parents exhibit strong in-group favoritism while childfree adults do not.

**Data Availability Statement:** Our replication materials including data and code are available at: https://osf.io/dp8tx/?view_only=0f13831daea349f982cff7835433178d.

## Introduction

Childfree individuals, who are also called 'childless by choice' or 'voluntarily childless', have decided they do not want biological or adopted children. Because childfree individuals do not desire children, they are uniquely different from several other groups of non-parents including 'not-yet-parents' who plan to have children in the future, 'childless' individuals who wanted to have children but could not have them due to infertility or life circumstances, 'undecided' individuals who are not sure if they want children in the future, and 'ambivalent' individuals who are not planning to have children in the future but are unsure whether they wanted to have children. Each of these groups of non-parents may have different characteristics, needs, and life experiences. Therefore, it is important to distinguish childfree individuals from other non-parents [1, 2].

Recent research suggests the childfree population is important to understand because its members have unique reproductive health [3] and end-of-life [4] needs. Childfree individuals also encounter challenges managing work-life balance [5–7] and with stereotypes [8–13]. Much of the emerging research on the childfree population either adopts a qualitative approach [5, 6, 14–18], or conducts quantitative analysis of a non-representative sample [9–11,

**Funding:** This work was supported by a grant from Michigan State University's Institute for Public Policy and Social Research to ZPN and JWN (http://ippsr.msu.edu/). The funders had no role in study design, data collection and analysis, decision to publish, or preparation of the manuscript.

**Competing interests:** The authors have declared that no competing interests exist.

13, 19, 20]. Although this prior work has provided insight, it does not allow conclusions to be drawn about population characteristics such as prevalence, age of decision, or interpersonal warmth judgements. Through a pre-registered direct replication of an earlier population-representative study [2], we aim to confirm earlier conclusions drawn about these characteristics of the childfree population.

Direct replication involves repeating "the critical elements (e.g., samples, procedures, measures) of an original study" with the intent "to evaluate the ability of a particular method to produce the same results upon repetition" [21]. Direct replication is valuable because it can help establish confidence that prior findings were not chance occurrences. Within the context of research on the childfree population, direct replication is important for several reasons. First, generalizable research on childfree individuals is still uncommon and direct replication can help increase our confidence in and understanding of the prevalence, age of decision, and interpersonal warmth judgements of this understudied population [1, 2, 22]. Second, direct replication can help rule out the possibility that findings related to childfree individuals were an artifact of sampling error or of particular current events (e.g., the delta wave of the COVID-19 pandemic). Third, direct replication can help ensure that new methods for measuring childfree individuals and other non-parents yield consistent, stable results.

Estimates of the prevalence of childfree adults in the United States population vary dramatically. Prior studies have estimated this prevalence at 1.3%–1.8% of ever-married women [23], 5%–9% of women aged 35–44 [24], 10%–20% of non-parent men aged 15–49 [25], 21.6% of all adults [2], and 44% of non-parent adults aged 18–49 [26]. Variations in study design make it difficult to compare these estimates or determine whether they are accurate. To increase confidence in the estimated contemporary prevalence of childfree adults and adults in other reproductive statuses, we aim to directly replicate one recent whole-population study [2]. Following that study's findings, we pre-registered six related hypotheses that the percent of Michigan adults who belong to each reproductive status group will not be statistically significantly different from previously [2] estimated values: 21.64% childfree (**H1a**), 49.62% parents (**H1b**), 9.58% not-yet-parents (**H1c**), 5.72% childless (**H1d**), 3.55% ambivalent (**H1e**), and 9.9% undecided (**H1f**). In addition to testing these confirmatory hypotheses, to gain insight into possible prevalence differences between demographic subgroups, we also conduct a series of exploratory analyses that compare the prevalence of childfree adults by sex, race, age, education, income, relationship status, and LGBTQIA identification.

Estimates concerning when adults decide to be childfree are similarly mixed. Earlier studies estimated that most childfree adults arrived at the decision late in life [27, 28]. In contrast, more recent studies estimate that most childfree adults are early deciders, arriving at the decision in the first several decades of life [2, 29, 30]. To increase confidence in the estimated age when contemporary adults decide to be childfree, we aim to directly replicate one recent study conducted on a population-representative sample [2]. Following that study's findings, we pre-registered six related hypotheses that the percent of Michigan adults who decided to be childfree in each decade of life will not be statistically significantly different from previously [2] estimated values: 3.6% before age 10 (**H2a**), 34.04% in their teens (**H2b**), 31.84% in their twenties (**H2c**), 17.14% in their thirties (**H2d**), 6.46% in their forties (**H2e**), and 6.91% after their forties (**H2f**). In addition to testing these confirmatory hypotheses, to gain insight into the validity of common responses to those deciding to be childfree, we also conduct a series of exploratory analyses to evaluate the evidence that childfree people change their mind or experience more life regret.

Prior research on attitudes toward childfree adults has been broadly consistent, pointing to others' negative feelings about this population. For example, compared to childfree raters, parents feel cooler toward childfree targets [1]. Similarly, childfree targets are viewed with

greater moral outrage than parents [10]. These past findings suggest that the childfree are a disliked outgroup. However, one recent study adopting a more complete rater-target evaluation of interpersonal warmth suggested the pattern is more qualified [2]. Specifically, it found that parents' warmth toward childfree adults was similar to childfree adults warmth toward other childfree adults, and toward parents. This implied that the apparent derrogation of childfree adults observed by earlier studies was illusory, and driven simply by a strong in-group favortism among parents. Despite this nuance, all these findings are broadly consistent with pronatalist (i.e. favoring parents and parenthood) norms observed most modern societies [31, 32]. To clarify and increase confidence in this finding, we aim to directly replicate it. Following that study's findings, we pre-registered three related hypotheses. First, we hypothesize that parents feel warmer toward parents than they feel toward childfree adults (**H3a**). Second, we hypothesize that parents feel warmer toward parents than childfree adults feel toward parents (**H3b**). Finally, we hypothesize that parents feel more ingroup warmth than childfree adults (**H3c**). In addition to testing these confirmatory hypotheses, to gain insight into the interpersonal warmth judgements of raters occupying other reproductive statuses, we also conduct a series of exploratory analyses.

## Methods

### Sample

Data used in our replication and exploratory analyses were collected between April 12 and April 22, 2022 as part of the State of the State Survey (SOSS), a recurring public opinion survey of Michigan adults conducted by the Institute for Public Policy and Social Research at Michigan State University. Because SOSS data is de-identified and publicly-available data, the Michigan State University Institutional Review Board determined them to be not 'human subjects' data (#STUDY00004613, 22 May 2020). Data collection occurred prior to the leak of the U.S. Supreme Court's draft decision on Dobbs v. Jackson Women's Health Organization on May 2, 2022. Therefore, our findings were not affected by the protests or legal changes related to this decision.

The SOSS dataset includes 1,000 Michigan adults who were matched on gender, age, race, and education to sampling frame based on the 2019 American Community Survey. The data includes sampling weights that were post-stratified on 2016 and 2020 Presidential vote choice, gender, age, race, and education. Table 1 reports the demographic characteristics of the unweighted and weighted sample.

### Pre-registered replication measures

**Reproductive status.** We used a series of up to three questions to classify respondents into six mutually-exclusive reproductive statuses. First, the SOSS asked "*Do you have, or have you ever had, any biological, step-, or adopted children?*". Respondents who answered "yes" to this question were classified as parents. Those who answered "no" to this first question were routed to a second question, "*Do you plan to have any biological or adopted children in the future?*". Respondents who answered "yes" were classified as not-yet-parents and those who answered "I don't know" were classified as undecided. Those who answered "no" to this second question were routed to a third question, "*Do you wish you had or could have biological or adopted children?*". Respondents who answered "yes" to this third question were classified as childless, those who answered "I don't know" were classified as ambivalent, and those who answered "no" were classified as childfree. The reproductive status of 20 respondents (2%) could not be determined due to missing data; these cases are dropped listwise.

**Table 1. Demographic characteristics of the unweighted and weighted sample.**

| Characteristic | Unweighted | Weighted |
|---|---|---|
| Sex | | |
| Men | 461 (46.1%) | 48.70% |
| Women | 539 (53.9%) | 51.30% |
| Race | | |
| White | 802 (80.2%) | 77.40% |
| Non-white | 198 (19.8%) | 22.60% |
| Education | | |
| Grad | 327 (32.7%) | 26.90% |
| Non-grad | 673 (67.3%) | 73.10% |
| Income | | |
| Over $60K | 414 (41.4%) | 39.70% |
| Under $60K | 586 (58.6%) | 60.30% |
| Relationship | | |
| Ever Partnered | 743 (74.4%) | 71.20% |
| Always Single | 255 (25.6%) | 28.80% |
| LGBTQIA Identification | | |
| Non-LGBTQIA | 879 (89%) | 90.30% |
| LGBTQIA | 109 (11%) | 9.70% |
| Age (mean) | 51.9 (sd = 17.2) | 50 (se = 0.7) |

**Age of decision.** We asked all respondents classified as childfree: "*How old were you when you decided you did not want to have children?*". To ease recall, we provided response options in decade intervals including "under 10", "10–19 years old", "20–29 years old", "30–39 years old", "40–49 years old", "50 or older", and "I don't know". The age of decision of 21 childfree respondents (9.7%) was not reported; these cases are dropped listwise.

**Interpersonal warmth.** We measured interpersonal warmth toward childfree individuals and parents using two feeling thermometer questions. These questions were presented in random order to participants to avoid order effects. To measure interpersonal warmth toward childfree individuals, we used the question: "*On a 0 to 100 scale, where 0 means very cold or unfavorable, and 100 means very warm or favorable, how do you feel toward people who never want to have or adopt children?*". To measure interpersonal warmth toward parents, we used the question: "*On a 0 to 100 scale, where 0 means very cold or unfavorable, and 100 means very warm or favorable, how do you feel toward people who have children?*". One or both of these warmth judgements was not reported by 47 respondents (6.2%); these cases are dropped listwise.

## Additional measures for exploratory analyses

In addition to our pre-registered replication, we conducted exploratory analyses using several additional demographic variables collected as part of the SOSS.

**Sex.** Respondents indicated their sex in response to the question, "*What is your sex?*", using one of the following response options: 'male', 'female', or 'intersex/other'. Because no respondents selected the 'intersex/other' option, we recoded sex as a binary variable to compare men and women.

**Race/Ethnicity.** Respondents indicated their race in response to the question, "*What is your race?*", using one or more of the following response options: 'White or Caucasian', 'African American or Black', 'Asian', 'Hawaiian or other Pacific Islander', 'American Indian or

Alaskan Native', or 'Other'. They also indicated their ethnicity in response to the question, "Are you of Hispanic, Latinx, or Spanish origin?" using the response options 'Yes' or 'No'. Because 93% of Michigan's population is either White alone or Black/African American, other racial and ethnic categories included a very small number of respondents. For this reason, and following US Census conventions, we recoded these two variables into a binary variable to compare respondents who are White alone and not Hispanic (i.e. White) to all others (i.e. Non-White).

**Age.**    To measure age, we first subtracted respondents' response to the question, "*In what year were you born?*" from 2022 (i.e., the year in which the data were collected). To facilitate comparisons between individuals in their prime childbearing years and individuals who were older, we recoded age as a binary variable indicating respondents were under 40 years old and respondents who were 40 and above.

**Education.**    Respondents indicated their educational attainment in response to the question "*What is the highest level of education you have completed?*", using ten categories. To facilitate comparisons between individuals, and following US Census conventions, we recoded education as a binary variable indicating whether respondents had completed a four-year college degree.

**Income.**    Respondents indicated their household income in response to the question "*Thinking about your household's total annual income from all sources (including your job) what was your family's annual income?*", using 12 unequally-spaced categories. Because the response options were not evenly spaced, and to facilitate comparisons between individuals, we recoded income as a binary variable indicating whether respondents had a household income that was above the 2022 Michigan median income of $60,000.

**Partnership status.**    Respondents indicated their partnership status in response to the question "*Are you currently married, divorced, separated, widowed, a member of an unmarried couple, or have you never been married?*", using seven categories. Whether or not one is childfree may depend on whether one has had the opportunity to be a parent, which may depend in part on whether one has (ever had) a partner. To facilitate comparing individuals, we recoded partnership status as a binary variable indicating whether respondents had ever been partnered, or were always single.

**LGBTQIA identification.**    Respondents indicated their LGBTQIA identification in response to the question "*Do you identify as Lesbian, Gay, Bisexual, Transgender, Queer, Intersex, or Asexual?*".

**Life regret.**    For reasons unrelated to this study, the State of the State survey included the five-item Satisfaction with Life Scale [33]. This scale includes an item that asks "If I could live my life over, I would change almost nothing," which respondents answer using a 7-point Likert scale ranging from 'strongly agree' (1) to 'strongly disagree' (7). Regret is defined, in part, as "a wish that [a mistake you made] could have been different or better" [34], therefore this item captures one facet of regret. We use responses to this item to measure individuals' regret over the choices they have made in their lives, where larger values reflect greater feelings of regret.

## Analysis plan

**Pre-registered replication.**    We pre-registered a direct replication of an earlier study on childfree adults on 9 August 2022 at https://osf.io/526dw. Our replication analyses are performed using the replication code provided by this study, and therefore exactly match the variable coding and model specification used in the original analysis. All reported estimates incorporate post-stratification sampling weights using the R survey package. Replication

data and materials for these analyses are available at https://osf.io/dp8tx.

**Exploratory analyses.** We also use these data to conduct a series of exploratory analyses that extend earlier findings concerning childfree adults.

First, after replicating prevalence findings, we separately estimate the prevalence of each reproductive status by population subgroups in terms of sex, race, age, education, income, partnership status, and LGBTQIA identification. To explore whether childfree adults are concentrated in specific segments of the population, we use t-tests to determine whether the prevalence of childfree adults differs between subgroups.

Second, after replicating age-to-decision findings, we explore two common responses to individuals making the decision to be childfree. To explore the response that they will change their mind, we examine the mean current age of childfree adults separately by their age at the time of decision. To explore the response that they will experience more life regret, we compare expressions of life regret by parents and childfree adults aged 70 or older.

Third, after replicating interpersonal warmth findings, we use regression analysis to examine whether findings hold controlling for other demographic variables. We also explore the interpersonal warmth felt toward both parents and childfree adults by individuals with other reproductive statuses. We present these as descriptive because, as the prevalence analyses confirm, these other statuses are rare and therefore associated samples are statistically underpowered for hypothesis testing.

All reported estimates incorporate post-stratification sampling weights using the R `survey` package. Replication data and materials for these analyses are available at https://osf.io/dp8tx.

## Results

### Prevalence of reproductive statuses

**Pre-registered replication.** Fig 1 shows the estimated prevalence of each reproductive status as a percent of the total adult population, with the associated 95% confidence intervals. We find that childfree adults comprise 20.94% (SE = 1.49, 95% CI: 18.03—23.86) of the adult population in Michigan. The prevalence of childfree adults is second only to parents who comprise 52.79% (SE = 1.91, 95% CI: 49.05—56.54) of the population. The other reproductive statuses are substantially less prevalent: Not-yet-parents (11.49%, SE = 1.43, 95% CI: 8.69—14.3), Undecided (7.44%, SE = 1.02, 95% CI: 5.45—9.43), Childless (4.27%, SE = 0.79, 95% CI: 2.72—5.82), and Ambivalent (3.07%, SE = 0.71, 95% CI: 1.68—4.46).

The prevalence estimates of parents, childfree, not-yet-parents, childless, and ambivalent adults are very similar to previously estimated values [2]. Additionally, their confidence intervals include, and therefore are not statistically significantly different from, the previously reported and hypothesized values. This supports hypotheses **H1a**—**H1e** and replicates prior findings about the prevalence of these reproductive statuses in the population.

We had also hypothesized that undecided adults comprise 9.9% of the adult population. However, in these data, we estimate that the prevalence of undecided adults is statistically significantly lower (7.44%). Therefore, we are unable to replicate an earlier prevalence estimate of undecided adults [2], and fail to support hypothesis **H1f**.

**Exploratory analyses.** Because the prior estimate of the prevalence of childfree adults in the population replicates, we can pool the two estimates to obtain a more precise estimate with a narrower confidence interval. A common effects meta-analysis model estimates the prevalence of childfree adults in the population is 21.35% (95% CI: 19.77%—23.01%).

To further explore the prevalence of childfree adults in the population, Fig 2 displays the estimated prevalence of each reproductive status within subgroups in terms of sex (A), race

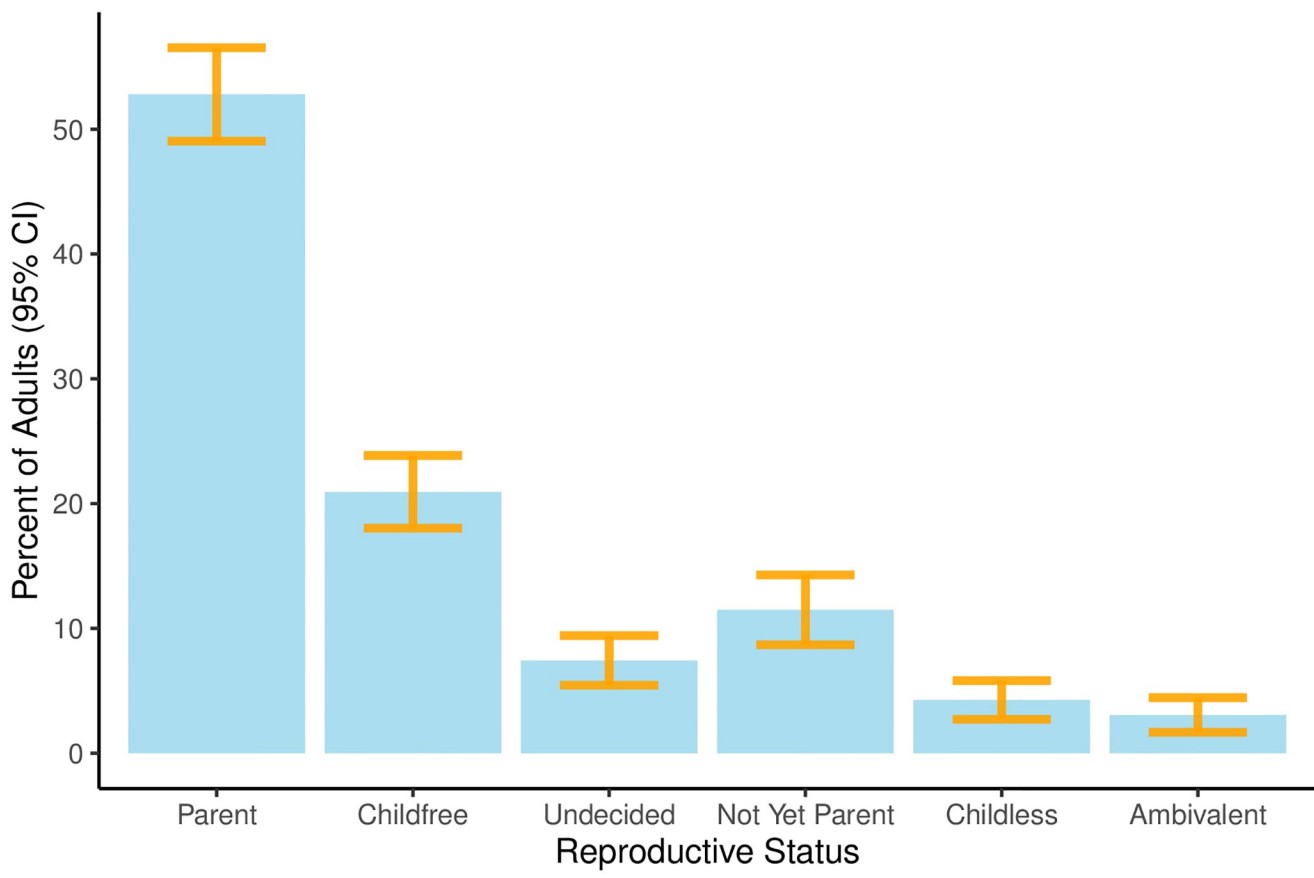

**Fig 1. Prevalence of reproductive statuses.**

(B), age (C), education (D), income (E), relationship status (F), and LGBTQIA identification (G; see S1 File for a tabular presentation). These analyses estimate the conditional probability (i.e. prevalence) of a reproductive status, given membership in a specific demographic subgroup: P(reproductive status | demographic characteristic). For example, panel A shows that 23.82% *of men* report being childfree, while only 18.2% *of women* report being childfree. Accordingly, the prevalences of the blue bars (one subgroup) in each panel sum to 100%, and likewise the prevalences of the orange bars (the other subgroup) in each panel sum to 100%. In this analysis, we focus on comparing the prevalence of being childfree between subgroups.

We observe no differences in the prevalence of childfree adults by age, education, or income (see S1 File for 4-category classifications of age and income). First, the percent of adults under 40 who are childfree (19.65%, SE = 2.96) is not statistically significantly different from the percent of adults age 40 or over who are childfree (21.48%, SE = 1.71; $\chi^2$ = 0.31, p = 0.58). Second, the percent of college graduates who are childfree (23.13%, SE = 2.45) is not statistically significantly different from the percent of non-graduates who are childfree (20.13%, SE = 1.82; $\chi^2$ = 1.04, p = 0.31). Finally, the percent of adults with above-median income who are childfree (18.17%, SE = 2.1) is not statistically significantly different from the percent with below-median income who are childfree (22.77%, SE = 2.04; $\chi^2$ = 2.82, p = 0.09).

In contrast, we do observe differences in the prevalence of childfree adults by sex, race, partnership status, and LGBTQIA identification. First, more men are childfree (23.82%, SE = 2.4)

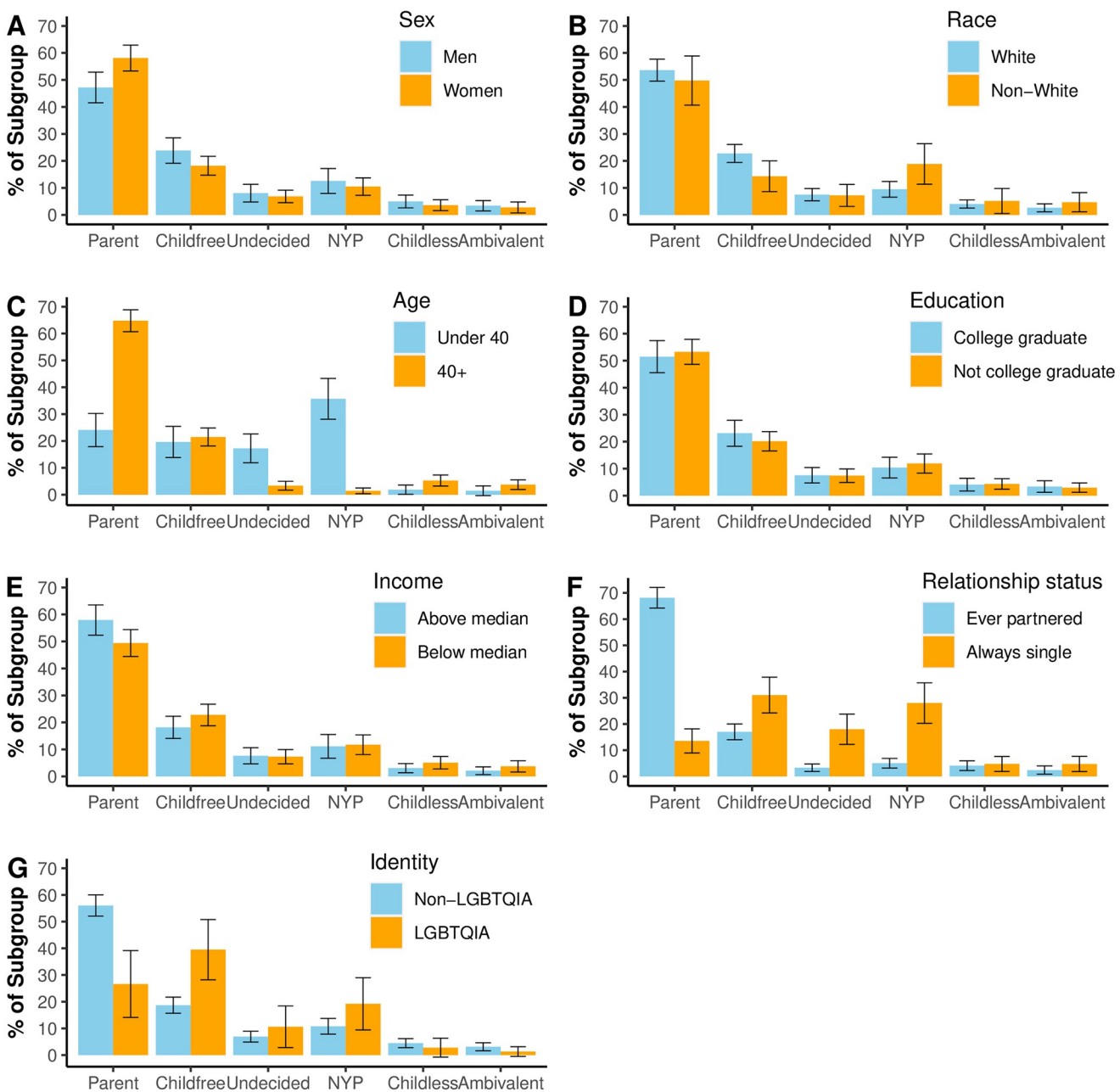

**Fig 2. Prevalence of reproductive statuses within demographic subgroups; P(reproductive status | demographic characteristic).**

than women (18.2%, SE = 1.79; $\chi^2$ = 4.53, p = 0.03). Second, more White adults are childfree (22.75%, SE = 1.7) than Non-White adults (14.32$, SE = 2.91; $\chi^2$ = 6.49, p = 0.01). Third, more adults who have always been single are childfree (31.03%, SE = 3.48) than adults who have ever been married or partnered (17%, SE = 1.54; $\chi^2$ = 22.02, p < 0.01). Finally, more adults who identify as LGBTQIA are childfree (39.48%, SE = 5.76) than adults who do not identify as LGBTQIA (18.69%, SE = 1.54; $\chi^2$ = 24.07, p < 0.01).

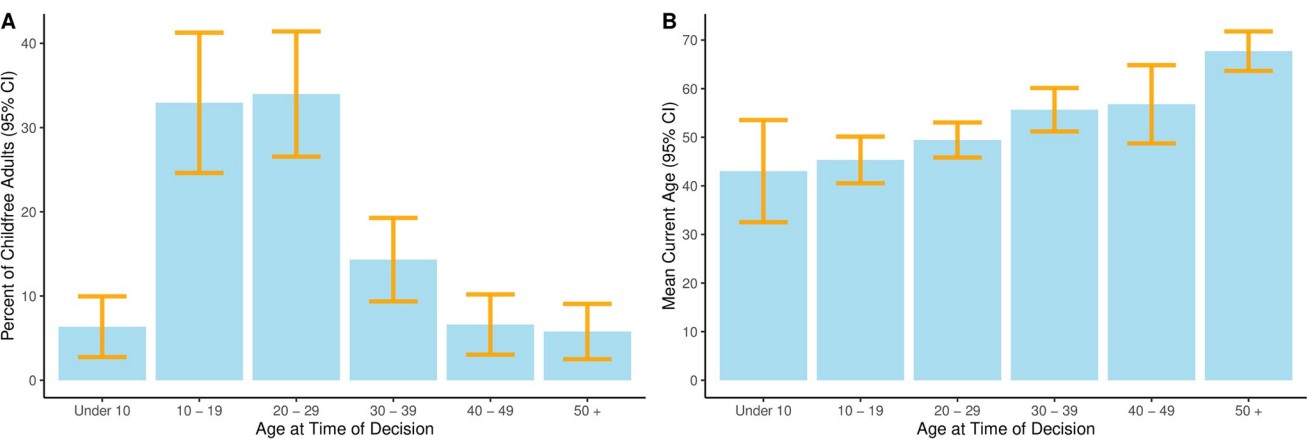

**Fig 3. (A) Age when childfree adults report that they decided to be childfree, (B) Mean current age of childfree adults, by age of decision.**

### Age of decision to be childfree

**Pre-registered replication.** Fig 3A shows the percent of the childfree population reporting that they decided they did not want children in each decade of life, with the associated 95% confidence intervals. We find that most childfree adults report that they decided they did not want children during prime childbearing years, in their teens (32.94%, SE = 4.25, 95% CI: 24.61—41.27) or twenties (33.99%, SE = 3.8, 95% CI: 26.55—41.42). Fewer childfree adults report that they arrived at this decision later in life, in their thirties (14.32%, SE = 2.53, 95% CI: 9.36—19.28), forties (6.61%, SE = 1.82, 95% CI: 3.04—10.19), or later (5.78%, SE = 1.68, 95% CI: 2.5—9.07), while a small percentage of childfree adults report that they knew before age 10 that they did not want children (5.78%, SE = 1.68, 95% CI: 2.5—9.07).

These estimated percentages are very similar to the previously estimated values [2]. Additionally, their confidence intervals include, and therefore are not statistically significantly different from, the previously reported and hypothesized values. This supports hypotheses **H2a**—**H2f** and replicates prior findings about when childfree adults report deciding to be childfree.

Although they replicate prior findings, these estimates must be interpreted with an important caveat. As cross-sectional data, they are subject to a potentially biasing truncation. Older childfree respondents could report deciding to be childfree at a young age or at an older age, while younger childfree respondents could *only* report deciding to be childfree at a young age. Therefore, the sample includes many people who could report an early decision, and fewer people who could report a late decision, which may account for the apparently high prevalence of early decisions. However, despite this caveat, these results closely mirror those from longitudinal studies using non-truncated data, which have found that a majority of permanently childless women arrived at their expectation of childlessness before age 30 [30].

**Exploratory analyses.** One common response to individuals who report not wanting children is that they will 'change their mind.' As a retrospective, cross-sectional survey, these data do not allow us to make inferences about whether individuals will change their mind, or whether they have changed their mind in the past. Moreover, survival bias presents a particular risk in analyzing the reported age-of-decision among currently childfree adults because only those who are still childfree at the time of the survey appear in the data, while formerly childfree adults who changed their mind do not.

In the absence of prospective longitudinal data, examining the current age of childfree adults can still provide some insight via a logical counterfactual. Suppose that people who decide early in life to be childfree frequently *do* change their mind, and eventually become parents (*p*). If this occurred, then it would mean childfree early-deciders observed in a cross-sectional survey would be relatively young (*q*) because many of the older formerly childfree early-deciders would have since changed their mind and would no longer be childfree. This counterfactual takes the logical form *If p, then q*. Fig 3B shows the mean current age (and 95% confidence interval) of childfree adults by the decade of life in which they reported deciding to be childfree. It shows a different pattern: childfree early-deciders are, on average, in their forties (i.e. *not q*). Specifically, we find that those who decided before their teens are now on average 43 years old (SE = 5.36), while those who decided in their teens are 45 (SE = 2.45), and those who decided in their twenties are 49 (SE = 1.84). This suggests that while some childfree adults may change their mind in the future, such mind-changing is not the dominant path (i.e. *therefore, not p* by *modus tollens*).

Another common response to childfree individuals is that they will experience regret about their lives. Again, without prospective longitudinal data we are unable to make inferences about childfree adults' future feelings of regret. However, we can examine whether parents and childfree adults in their late years of life express different levels of life regret. Focusing on adults aged 70 or older, we find that parents express more life regret (M = 3.87, SE = 0.20) than childfree adults (M = 3.30, SE = 0.39), but that the difference is not statistically significant ($t_{127}$ = 1.29, p = 0.20). This suggests that childfree adults do not experience more life regret than parents in their late years of life.

## Interpersonal warmth

**Pre-registered replication.** Fig 4A summarizes the mean interpersonal warmth judgements of parents (dashed red line) and childfree adults (solid blue line), with the associated 95% confidence intervals.

First, we find that parents feel significantly warmer toward parents (M = 82.99, SE = 1.00) than toward childfree adults (M = 67.67, SE = 1.53; t(511) = -9.56, p < 0.001). This supports hypothesis **H3a** and replicates prior findings of ingroup favoritism among parents.

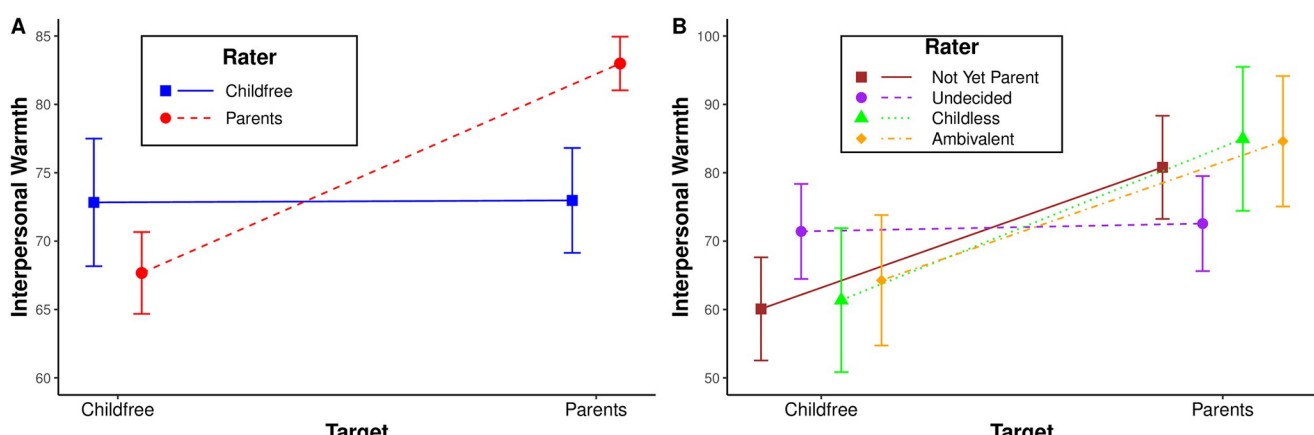

**Fig 4. (A) Interpersonal warmth felt by childfree adults and parents toward each other, (B) Interpersonal warmth felt toward childfree adults and parents by others.**

Second, we find that parents feel significantly warmer toward parents than childfree adults feel toward parents (M = 72.98, SE = 1.95, t(709) = -4.55, p = < 0.001). This supports hypothesis **H3b** and replicates prior findings that people are polarized in their feelings toward parents.

Finally, we find that parents feel more warmth toward parents than childfree people feel toward childfree people (M = 72.83, SE = 2.38, t(709) = -3.93, p = <0 0.001). This supports hypothesis **H3c** and replicates prior findings that group differences in interpersonal warmth are driven by parents' ingroup favoritism.

**Exploratory analyses.**   The study we replicate examined the mean interpersonal warmth felt by childfree adults and parents toward each other [2]. To further explore these patterns, Table 2 reports the results of a regression in which we predict how much warmer respondents feel toward parents than toward childfree adults (i.e., DV = warmth toward parents—warmth toward childfree adults) as a function of their their status as a parent or childfree adult and the same demographic characteristics examined in Fig 2. Confirming the finding illustrated by the steep red line and flat blue line in Fig 4A, we find that parents feel statistically significantly warmer toward parents than toward childfree adults ($b$ = 13.142, $se$ = 3.122, $p$ < 0.001) even after controlling for demographic characteristics. We also observe that women feel statistically significantly cooler toward parents than toward childfree adults ($b$ = −8.989, $se$ = 2.684, $p$ = 0.001). This suggests that the ingroup favoritism observed among parents may have gendered dimensions that are worth exploring further.

The replication and exploratory analyses reported above focus only on childfree and parent raters. Fig 4B extends this analysis by examining the interpersonal warmth felt toward childfree adults and parents by those who are not-yet-parents (solid brown), undecided (dashed purple), childless (dotted green), and ambivalent (dotdashed orange). Because these other groups are rare in the population (see Fig 1), this exploratory analysis lacks the statistical power to test for differences in warmth judgements, but broad patterns are still evident. First, the warmth judgements of undecided adults are most similar to childfree adults: they feel similarly warm toward both childfree adults and parents. This might be expected because undecided adults are unsure which of these two groups they wish to join. Second, the warmth judgements of other groups are similar to those of parents: they feel cooler toward childfree adults than toward parents. This might be expected for not-yet-parents and childless adults because these groups want(ed) children and aspire(d) to be parents. Finally, childless adults feel the most different toward childfree adults (M = 61.36, SE = 5.37) and parents (M = 84.97, SE = 5.37), perhaps reflecting their disapproval of those who voluntarily choose not to have children and their admiration of those who were able to have children.

**Table 2. Difference in warmth felt toward parents and childfree adults, as a function of parent/childfree status and demographic characteristics.**

| Variable | b | se | t | p |
|---|---|---|---|---|
| Intercept | 9.054 | 4.926 | 1.838 | 0.067 |
| Parent | 13.142 | 3.122 | 4.210 | < 0.001 |
| Woman | -8.989 | 2.684 | -3.350 | 0.001 |
| White | -3.549 | 4.345 | -0.817 | 0.414 |
| LGBTQIA | -2.699 | 8.307 | -0.325 | 0.745 |
| Ever Partnered | 3.767 | 3.228 | 1.167 | 0.244 |
| Under 40 | -3.294 | 3.347 | -0.984 | 0.325 |
| College Graduate | -3.986 | 2.484 | -1.605 | 0.109 |
| Above Median Income | -1.866 | 2.443 | -0.764 | 0.445 |

## Discussion

In this study, we directly replicated nearly all prior findings related to the prevalence, age of decision and interpersonal warmth judgements of childfree adults reported in recent previous research [2]. Specifically, our study provides confirmatory evidence that childfree individuals are numerous, comprising over one in five Michigan adults, and tend to be early-deciders who come to their decision during their teens and twenties [27]. Furthermore, we replicated prior evidence of ingroup favoritism in the interpersonal warmth judgements of parents. Our ability to directly replicate prior findings suggests that they cannot be attributed to an anomalous sample, to fleeting changes in respondents' views on having children, or to other contextual factors such as the COVID-19 pandemic. Because these methods yield replicable findings in Michigan, a wider-scope study is warranted to determine whether the estimated prevalence, age of decision, and interpersonal warmth judgements of childfree adults generalize to other regions of the United States and to other countries.

The only estimate from previous research that we did not replicate is the prevalence of adults who are undecided about having children (**H1f**). Specifically, the percentage of undecided adults in the current study was significantly lower than in previous research [2]. Perhaps some adults who were undecided in 2021, which was marked by significant uncertainty due to the COVID-19 pandemic, had made a decision about their reproductive plans by 2022. However, the fact that we replicate prior prevalence estimates of all other reproductive statuses suggests that these 2022-deciders did not settle on the same reproductive path (e.g., they did not all decide they wanted children, and thus become not-yet-parents).

Our exploratory analyses of the prevalence of childfree adults by subgroup offer insight into who is more likely to decide to be childfree. Media narratives often portray childfree adults as young millennials who forgo parenthood due to a desire for higher educational attainment or a lack of economic resources [35–37]. However, contrary to these narratives, we found no differences in the prevalence of childfree adults by age, education, or income. Instead, we revealed differences in the prevalence of childfree adults by sex, race, partnership status, and LGBTQIA identification. First, men are more likely than women to identify as childfree. Women may be more hesitant to disclose a childfree identification due to more intense pronatalist pressures surrounding motherhood [32]. Alternatively, men may simply be more likely than women to adopt a childfree lifestyle. Indeed, recent demographic research suggests the percentage of men who do not want children has increased in the past 20 years [25]. Second, White adults are more likely to identify as childfree than non-White adults. More research is needed to understand the interplay between race and reproductive decision-making. Finally, adults who have always been single and adults who identify as LGBTQIA are more likely to be childfree. These individuals are more likely to reject traditional definitions of family based solely on biological relations [38, 39] and face increased barriers to biological or adoptive parenthood [40–42] that may contribute to their decision to be childfree.

Our exploratory analyses of age of decision suggest that common responses to childfree adults lack merit [18, 43]. Specifically, although childfree adults are often told that they will 'change their mind,' we found that early-deciders were on average in their forties, suggesting a pattern of persistence in their decision to be childfree. Additionally, although childfree adults are often told that they will later 'regret their lives,' those who were 70 or older were no more likely to express feelings of life regret than their parent counterparts. Despite a lack of evidence to support responses that childfree people will 'change their minds' or 'regret their lives,' these responses continue to be ubiquitous and can have negative consequences for childfree adults. For example, childfree adults often report feeling stigmatized and dismissed by others [15, 32, 43, 44]. Additionally, medical providers routinely deny childfree adults' access to voluntary

sterilization based on beliefs that they will change their mind or experience life regret [45–47]. Therefore, more outreach and education are necessary to dismantle and reduce these responses to childfree individuals.

Our exploratory analyses of interpersonal warmth judgements provide an understanding of how other adults without children view parents and childfree adults. Notably, undecided adults who do not know if they want children have a pattern of interpersonal warmth judgments that mirrors those of childfree adults: they are similarly warm to parents and childfree adults. This signals that undecided adults may recognize and be open to both childfree and parent lifestyles. However, longitudinal research is necessary to track how initially undecided adults wind up in terminal reproductive statuses. Not-yet-parents, childless adults, and ambivalent adults all have a pattern of interpersonal warmth judgments that mirrors those of parents. That is, they are more warm toward parents than childfree adults. For both not-yet-parents and childless adults, this signals that they aspire(d) to be, and therefore esteem, parents. Such favoritism toward parents, even among many non-parents, may place childfree adults at increased risk of stigmatization, othering, social exclusion, and discriminatory practices [15, 32, 43, 44, 48, 49].

The current study contributes to our growing understanding of childfree adults through a pre-registered direct replication of prior research and new exploratory analyses in a large, representative sample. Nevertheless, some limitations should inform the interpretation of the results. First, like the previous research we replicated [2], we used data that were limited to Michigan adults. Although the demographics of Michigan's adult population are similar to the demographics of the U.S. adult population, research is still needed to determine whether these findings would generalize nationally and internationally. Second, we used cross-sectional data that cannot provide information about the developmental trajectories that led childfree adults to their decision, and cannot identify formerly childfree adults who later became parents. Longitudinal research using panel data could help determine how adults dynamically shift in and out of different reproductive statuses over time [30]. Third, consistent with the study we were replicating, we adopted the term "reproductive status" to describe our categorization of parents and different groups of non-parents [2]. However, because our operational definition of parents includes both biological and non-biological (e.g., adoptive or step) parents, the term "parental status" may be more appropriate to use as a description of this categorization in future studies.

Although childfree adults are a distinct population with unique healthcare [3] and workplace needs [5–7], there is still limited generalizable research on their prevalence, age of decision, and interpersonal warmth judgements. Replicating past findings [2], our study provides additional confidence that childfree adults are numerous, tend to make the decision to not have children early in life, and that parents exhibit strong in-group favoritism while childfree adults do not. Additionally, our exploratory analyses provide a more nuanced understanding of who is likely to be childfree and dispel common responses to childfree decisions. Given the large number of adults who identify as childfree, it will be important to conduct further large-scale generalizable studies of this population.

## Supporting information

**S1 File.**
(PDF)

## Author Contributions

**Conceptualization:** Jennifer Watling Neal, Zachary P. Neal.

**Formal analysis:** Jennifer Watling Neal, Zachary P. Neal.

**Investigation:** Jennifer Watling Neal, Zachary P. Neal.

**Methodology:** Jennifer Watling Neal, Zachary P. Neal.

**Writing – original draft:** Jennifer Watling Neal, Zachary P. Neal.

**Writing – review & editing:** Jennifer Watling Neal, Zachary P. Neal.

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
