## [Decision Letter · Decision Letter 0]

4 Jan 2023

PONE-D-22-26908Prevalence, age of decision, and interpersonal warmth judgements of childfree adults: Replication and extensionsPLOS ONE

Dear Dr. Watling Neal,

Thank you for submitting your manuscript to PLOS ONE. After careful consideration, we feel that it has merit but does not fully meet PLOS ONE’s publication criteria as it currently stands. Therefore, we invite you to submit a revised version of the manuscript that addresses the points raised during the review process.

Both reviewers and I also agree that the manuscript explores an important and understudied topic. There was some disagreement as to whether a major or minor revision is needed for the manuscript.  In the revised manuscript, please be sure to detail your responses to the reviewers as to what changes were made and why. Please ensure that there is a sound scientific rationale for the replication and the text justification is sufficient for readers to ascertain this (see reviewer 1 comments). All comments are below for your reference.

We look forward to receiving your revised manuscript.

Kind regards,

Janna Metzler

Academic Editor

PLOS ONE

Journal Requirements:

Reviewers' comments:

Reviewer's Responses to Questions

**Comments to the Author**

1. Is the manuscript technically sound, and do the data support the conclusions?

Reviewer #1: Partly

Reviewer #2: Yes

2. Has the statistical analysis been performed appropriately and rigorously? 

Reviewer #1: Yes

Reviewer #2: Yes

3. Have the authors made all data underlying the findings in their manuscript fully available?

Reviewer #1: Yes

Reviewer #2: Yes

4. Is the manuscript presented in an intelligible fashion and written in standard English?

Reviewer #1: Yes

Reviewer #2: Yes

5. Review Comments to the Author

Reviewer #1: This paper uses data from a representative survey of adults in Michigan to estimate the prevalence of remaining childfree as well as the reported age of the decision and judgments of interpersonal warmth. The manuscript is framed as a replication, using 2022 data to re-estimate population estimates from an initial 2021 survey (with an earlier study in 2020 as well).

The authors estimate that about 21-23% of adults are childfree – i.e., have chosen to never have children. This estimate is higher than previous work, and it is based on desires and plans rather than fecundity (i.e., ability to have children), providing a valuable perspective on remaining childfree by choice. The data also allow for distinguishing those who are not-yet-parents, childless (unable to have children), undecided, or ambivalent. The authors provide interesting exploratory analyses of the characteristics of adults across different groups of non-parents and comparisons of perceived warmth of childfree adults and parents.

This is an interesting paper on a timely topic. However, there are some weaknesses in its conceptualization and methods that could be strengthened.

1. Replication

In various social sciences, replication has been most valuable when it has tested a theoretical proposition (including its stability or robustness across data sets or methods) or when it has been employed technically to test the validity and reliability of a method or measure. For the current manuscript, it is unclear whether its framing as a replication is clearly justified.

The current study’s hypotheses are essentially population estimates, empirically informed by a prior study by the same researchers. From a technical view, the sampling and survey methods in both studies appear to be the same (though the current manuscript should provide more information to assess this). Both studies use representative samples of Michigan residents (in 2021 and 2022, respectively). Thus, probability theory would predict that the measured estimates should be close (within a given margin of error). That is indeed the finding of the current manuscript – nearly all estimates are within the confidence intervals of the initial study (except for the undecided, whose prevalence is about 2.5 percentage points lower).

It seems that a replication in this case may be unwarranted unless there is reason to question whether the methods are indeed the same. The consistency in the proportions of people in each group is a useful technical verification that may be more appropriately included as a technical note for a larger, more meaningful set of analyses.

2. Extending the analyses

The repeated data collection allows for greater certainty in the estimates, and the consistency in the methods offers the opportunity for deeper exploration of the data. Pooling the data from the two surveys could allow for increased sample size and statistical power for a more detailed examination of the characteristics of people in each group. For example, the authors should move beyond the binary coding of sample characteristics (e.g., education, income, race-ethnicity) to examine more detailed categories.

Alternatively, a larger sample size could allow for multivariate analyses to expand on the current bivariate examinations of the sample characteristics. For example, a multinomial logistic regression method could allow for exploration of factors that are related to being childfree compared to a parent, not-yet-parent, undecided, etc. (e.g., are differences still significant after controlling for other factors?).

3. Use of cross-sectional data

Conceptually, the manuscript could be strengthened by providing a stronger justification for reporting results from cross-sectional data for all adult ages. Given the emphasis in prior work (especially fertility studies) on life course processes, why is it important to provide estimates of the proportions of childfree adults in the population (i.e., a cross-section)?

4. Attention to age and life course processes

The examination of reported ages of making the choice to remain childfree raises some problematic questions. Earlier work by demographers using longitudinal data has reported a dynamic process in decisions, whereby people adjust their desires and preferences in response to both individual and larger circumstances (e.g., see Gemmill, 2019).

With this in mind, the analyses of the reported ages of childfree decisions could be more specific and refined with regard to patterns that vary by age. In particular, Figure 3 reports the average ages of people reporting each 10-year age category of when they decided not to have children. But the cross-sectional nature of the data means that younger respondents are reporting on a shorter/younger potential range of ages for making this decision (i.e., their experiences are truncated). It is therefore not surprising (and an artifact of the data) that a higher proportion of people reported younger ages for their decisions – there is a much larger proportion of people in the sample who experienced the younger ages, whereas only the older respondents have the possibility of reporting decisions at older ages. Also, the decisions and their meanings are likely to vary with regard to one's position in the reproductive ages (e.g., they may become more acute as one approaches the end of the "biological clock"). In general, this section was least clear in its conceptualization and data presentation and less convincing in its interpretations.

5. Other points

- The distinction between “undecided” and “ambivalent” is unclear, and the text itself is inconsistent in defining these groups. From my reading, the “undecided” report that they don’t know if they plan to have children, whereas the “ambivalent” report that they don’t plan to have children but don’t know if they wish they could have (had) children. It would be helpful to make these distinctions more clearly in the writing and also provide some explanation or justification of why this distinction might be important. Importantly, how are these groups conceptually different one another and from the “childfree”?

- The numbers in Figure 2 could be better presented with a table. This would allow for clearer reporting and easier comparison of the characteristics of people across the parental status groups (as well as indication of statistically significant differences).

- On page 9 (line 278-279), it is unclear what the authors mean by the statement that “examining the current age of childfree adults can still provide some insight via a counterfactual.” The cross-sectional data do not allow for observation of whether people have changed their minds (and prior research using longitudinal data shows that decisions do change over time). Indeed, the retrospective accounts of age of deciding to remain childfree may be more precisely seen as current, subjective assessments rather than accurate factual reports of past events. Further, since only childfree adults were asked this question, we do not know anything about the subjective reports of ages and decisions among people in the other groups, some of whom may have been “childfree” at an earlier age but are now parents or are childless, undecided, or ambivalent.

- The attempt to consider regret is interesting and laudable, but it is unclear whether the available measure if appropriate. The question asked, “If I could live my life over, I would change almost nothing.” The word “regret” is not actually in this question, but is inferred by the researchers. Further, the question is so broad (there is no referent or target specified for what one might change about their life) that its use to infer regret about parental status seems unjustified and unsupported by the actual data. It is also unclear how adults at very different ages might interpret this question.

- The descriptive characteristics are described only in terms of a simplistic, binary coding. This section would be more informative by providing more detail about the sample characteristics and how they vary for people in the different parental status groups.

- A small note – the authors use the phrase “reproductive status” to describe the conceptual groups at the heart of this study, but it seems that “parental status” might be more appropriate, especially since the survey questions ask about both biological and adoptive parenthood.

- The results on perceptions of interpersonal warmth are strong and interesting. Here, using pooled data might allow for more statistical power in the comparisons. Also, multivariate analyses would allow for assessment of the robustness of these results after comparing for other factors.

Overall, an interesting analysis of a unique data source. The study provides thoughtful examination of different stances with regard to not having children, including perceptions of people in different parental status groups.

Reference:

Gemmill A. (2019). From some to none? Fertility expectation dynamics of permanently childless women. Demography, 56(1), 129–149.

Reviewer #2: PONE-D-22-26908

Prevalence, age of decision, and interpersonal warmth judgements of childfree adults: Replication and extensions

The goal of this study is to provide a replication and extension of a prior study of the childfree population in the US, using a sample of people in Michigan as a proxy. The authors state:

“To clarify these characteristics of the contemporary childfree population, we conduct a pre-registered direct replication of a recent population-representative study. All estimates concerning childfree adults replicate, boosting confidence in earlier conclusions that childfree people are numerous and decide early in life, and that parents exhibit strong in-group favoritism.”

The study is well designed, explores an important and understudied topic, and uses appropriate analytical methods. The paper is easy to read, well organized, and mostly clear. The figures show the results in a way that is easy for the ready to follow. That the hypotheses were pre-registered and that data and syntax are available adds confidence to the rigor of the results.

Below I summarize the particular strengths of the manuscript and make some minor suggestions for improvement.

Strengths

*The topic is important

*replication and clarification of the childfree population is worthwhile

*in addition to practical concerns (e.g. ensuring that CF are included in family research and planning for the needs of people who are aging without adult child caregivers), there are theoretical reasons to study those who make counter normative decisions - and to question if norms are changing.

*The three questions designed to classify participants as parents, involuntary childless, not yet parents, unsure, and childfree have face validity. It might be useful to think a little more about what “not wanting to have children” really means. People could not want children because they cannot afford them, because they do not want to pass down a genetic abnormality, because they think that doing so is a problem for environmental sustainability, for some other reason, or because they simply enjoy not being parents. Do these different reasons matter for being considered “childfree”?

Finding that about 20% of adults in Michigan are childfree is very interesting and I suspect larger than most Americans would expect. That the percentage childfree does not differ by income and education is very interesting; it suggests that people are not making this decision based on financial concerns.

The findings about differences by sex, race, partnership status and LGBTQIA identification are so interesting and create great curiosity (at least in me!) to learn what explains these patterns. Also fascinating is the early age at which people decided they were childfree. I do wonder if people would call themselves childfree.

The lack of a difference in level of regret between parents and the childfree is so interesting. It does seem as if some people who do not want children worry that they will regret their choice, and this finding is potentially useful information.

Suggestions for improvement

The claim that it is hard to identify the childfree in existing datasets seems unsupported unless the definition requires that people claim the label “childfree”. Identifying people who are not parents is generally easy in large studies such as the National Survey of Family Growth. It is also possible to determine if people do not intend children. What is harder is determining how free people are to chose to not have children (e.g. is it because they do not want them, or because of cost or health issues?).

*It was surprising in the abstract to have the last phrase focus on “parents” and their in-group favoritism as the childfree, not parents, seems to be the focus of the paper. Is it more accurate to say that the childfree perceive that parents exhibit strong in-group favoritism?

“Interpersonal warmth judgements” is less obvious as an issue than the other topics (e.g. prevalence and age of decision). Perhaps mention the justification for studying “interpersonal warmth” in the introduction? The phrase seems a bit out of place.

It might be useful to state with the hypotheses that the comparison will be based upon a p-value confidence interval because the hypotheses values are so precise (to the 2 decimal place).

The replication seems to also confirm the relative stability of these estimates over time, is this correct?

The sentence below seems to have a typo:

“In addition to testing these confirmatory hypotheses, to gain insight into the validity45of common responses to those deciding to be childfree, we also conduct a series of46exploratory analyses we evaluate the evidence that childfree people change their mind47or regret their decision”

I am not following logic of these sentences on page 8: “Specifically, it found that parents’ warmth toward childfree55adults was similar to childfree adults warmth toward other childfree adults, and56toward parents. This implied that the apparent derrogation of childfree adults57observed by earlier studies was illusory, and driven simply by a strong in-group58favortism among parents”

If parents had higher positive ratings towards other parents and lower ratings towards the childfree, and the childfree had higher positive ratings towards other childfree folks and lower ratings towards parents, then each group would be demonstrating in-group preference. The statement above suggests that parents had the same warmth towards the childfree as towards other parents. Am I misreading the sentences? Figure 4 suggests to me that among parents, there is much lower warmth towards the childfree and much higher warmth toward their fellow parents, but among the childfree, warmth is similar towards their fellow childfree and towards parents. This latter finding is consistent with the idea that only parents seem to have ingroup favoritism.

I am trying to figure out why the hypotheses suggest higher in-group favoritism among parents than among the childfree.

It would be helpful to have some theory and not just prior research in order to understand why people would have negative attitudes towards the childfree.

The following phrase on page 12: “...and are subject to lower levels of421interpersonal warmth from parents due to parents’ ingroup favoritism.” would be more clear if “from parents” was change to “from people who are parents” because it is unclear if the reference is their own parents or parents in general (the parents of the childfree).

It is possible that the “undecided” might be waiting for a partner to decide if they will have a child or not.

6. PLOS authors have the option to publish the peer review history of their article (what does this mean?). If published, this will include your full peer review and any attached files.

Reviewer #1: No

Reviewer #2: No

---

## [Decision Letter · Decision Letter 1]

6 Mar 2023

Prevalence, age of decision, and interpersonal warmth judgements of childfree adults: Replication and extensions

PONE-D-22-26908R1

Dear Dr. Watling Neal,

We’re pleased to inform you that your manuscript has been judged scientifically suitable for publication and will be formally accepted for publication once it meets all outstanding technical requirements.

Kind regards,

Janna Metzler

Academic Editor

PLOS ONE

Additional Editor Comments (optional):

Reviewers' comments:

Reviewer's Responses to Questions

**Comments to the Author**

1. If the authors have adequately addressed your comments raised in a previous round of review and you feel that this manuscript is now acceptable for publication, you may indicate that here to bypass the “Comments to the Author” section, enter your conflict of interest statement in the “Confidential to Editor” section, and submit your "Accept" recommendation.

Reviewer #2: All comments have been addressed

2. Is the manuscript technically sound, and do the data support the conclusions?

Reviewer #2: (No Response)

3. Has the statistical analysis been performed appropriately and rigorously? 

Reviewer #2: (No Response)

4. Have the authors made all data underlying the findings in their manuscript fully available?

Reviewer #2: (No Response)

5. Is the manuscript presented in an intelligible fashion and written in standard English?

Reviewer #2: (No Response)

6. Review Comments to the Author

Reviewer #2: (No Response)

7. PLOS authors have the option to publish the peer review history of their article (what does this mean?). If published, this will include your full peer review and any attached files.

Reviewer #2: No

---

## [Editor Report · Acceptance letter]

14 Mar 2023

PONE-D-22-26908R1 

Prevalence, age of decision, and interpersonal warmth judgements of childfree adults: Replication and extensions 

Dear Dr. Watling Neal:

I'm pleased to inform you that your manuscript has been deemed suitable for publication in PLOS ONE. Congratulations! Your manuscript is now with our production department. 

Kind regards, 

on behalf of

Dr Janna Metzler 

Academic Editor

PLOS ONE